# The Effect of Fe Content on the Microstructure and Tensile Properties of Friction-Stir-Welded Joints in Recycled Cast Aluminum Alloy

**DOI:** 10.3390/ma17010064

**Published:** 2023-12-22

**Authors:** Fujian Gong, Gang Feng, Yajun Wang, Sha Lan, Jinsheng Zhang, Chun Wang, Jianhua Zhao, Qing Yang, Zhibai Wang

**Affiliations:** 1College of Material Science and Engineering, Chongqing University, Chongqing 400045, China; gfjgfj2023@163.com (F.G.); shalan2023@163.com (S.L.); wangchun412@outlook.com (C.W.); 2Changan Automobile Co., Ltd., Chongqing 401120, China; fenggang202305@163.com (G.F.); zhangjs2023@outlook.com (J.Z.); yangqing202305@outlook.com (Q.Y.); wangzhibai2023@outlook.com (Z.W.); 3National Engineering Research Center for Magnesium Alloys, Chongqing University, Chongqing 400044, China

**Keywords:** Fe content, friction-stir-welded joint, recycled aluminum alloy, mechanical properties

## Abstract

The presence of the impurity element Fe significantly influences the overall performance of recycled aluminum alloy. This study aims to elucidate the impact of Fe content on the microstructure and tensile properties of friction-stir-welded (FSW) joints in recycled cast A356 aluminum alloy. Three samples with varying Fe content were prepared for FSW joints. The quality of the weld zone was meticulously assessed through macrostructure and microstructure analyses. The tensile strengths of the joints were carefully evaluated and correlated with the microhardness and microstructure of the weld zone. The research findings reveal that, among the three fabricated joints, the one with an Fe content of 0.3 wt.% demonstrates the most favorable tensile performance. This particular joint exhibits the highest tensile strength of 153 MPa, commendable yield strength of 90 MPa, and a favorable elongation of 5.7%. The mechanisms responsible for grain refinement in the weld nugget zone involve plastic deformation and dynamic recrystallization. Significantly, the disruptive effects of friction-stir action on eutectic silicon phases and rich iron phases emerge as crucial factors contributing to the enhanced performance of the weld nugget zone in the welded joint.

## 1. Introduction

Aluminum alloys have widespread applications for structural components in electrical, automobile, aerospace, and general engineering industries owing to their remarkable castability, corrosion resistance, and notably, a high strength-to-weight ratio in the heat-treated condition [1,2]. Among the myriad of aluminum alloys available, A356 stands out as a popular casting alloy widely employed in automotive components like engine blocks, cylinder heads, and suspension parts [3,4,5]. The low impurity content and stable composition of primary aluminum alloys ensure outstanding performance; however, it is essential to acknowledge the environmental challenges associated with the production of primary aluminum. The electrolysis process involved is known for its energy intensity and substantial CO_2_ emissions, posing a potential hurdle to the environmental advantages of utilizing aluminum components in lightweight vehicles. In contrast, recycling aluminum emerges as a significantly more economical and environmentally friendly alternative, requiring only a fraction of the energy needed for primary production. Despite these benefits, challenges arise in the recycling of aluminum alloys, particularly in the control of impurity elements during the recycling process.

Iron is readily absorbed during the melting process, inevitably present in commercial-cast aluminum–silicon–magnesium alloys. The presence of iron in these alloys can result in the formation of intermetallic compounds, notably Fe-rich plate-like particles. These particles can exert a significant influence on the mechanical properties of the castings [6,7,8]. These intermetallic compounds can cause high stress concentration, thereby promoting crack initiation and reducing the ductility of the castings [9,10,11]. It is widely acknowledged that the presence of iron in aluminum alloys can have detrimental effects on casting performance, mechanical properties, corrosion resistance, and other key aspects. Consequently, in practical production settings, mitigating the adverse impact of iron-rich phases in aluminum alloys primarily involves employing two distinct methods. Firstly, the iron content in the aluminum alloy melt can be directly reduced through various means, such as gravity sedimentation, electromagnetic separation, and centrifugal iron removal. This approach aims to minimize the iron concentration in the alloy, thereby enhancing its overall performance. The second method involves modifying the morphology of the iron-rich phase within the alloy. This can be achieved by incorporating alloying elements or subjecting the melt to specific heat treatments. Among these, heat treatment plays a crucial role in transforming the Fe-rich phase. This process typically includes steps such as dissolving, crushing, and spheroidizing the iron-rich phase, thereby altering its morphology for improved alloy characteristics. Liu et al. [12] proposed a model of dissolution, fragmentation, and spheroidization of the acicular β-AlFeSi phase during heat treatment. Selecting an appropriate heat treatment temperature is the key factor to ensure that the iron-phase morphology can be improved without causing excessive burning loss of castings. Friction stir treatment of aluminum alloy is also a means to change the appearance of the harmful iron-rich phase. In the process of friction stir welding (FSW), the friction heat generated by tool tip rotation and mechanical deformation can achieve the effect of changing the morphology of the Fe-rich phase.

The fusion-stir-welding process for cast aluminum alloys frequently encounters difficulties stemming from the presence of typical flaws in the welds, including porosity, oxide inclusions, and hot cracking, all of which substantially impair the joint’s overall integrity. As an efficacious welding approach, friction stir welding has demonstrated its ability to prevent the manifestation of the aforementioned defects in aluminum alloys [13,14,15,16]. In a study conducted by Santella et al. [17], it was discovered that friction stir processing of the A356 aluminum alloy brought about notable improvements. Their findings revealed a reduction in alloy porosity and a more uniform distribution of second-phase particles following the application of this processing technique. Additionally, the removal of dendritic microstructures in the stir zone led to the enhancement of mechanical properties, particularly in terms of increased ultimate tensile strength, ductility, and fatigue life for both alloys. Additionally, many studies [18,19,20,21] have shown that performing friction stir processing on aluminum alloy leads to grain refinement in the stir zone and improves the microstructure morphology, particularly the fracturing of dendrites. As a result, the tensile strength, hardness, and elongation of the alloy’s microstructure in the stir zone are enhanced; however, there is limited research on the influence of Fe content on the friction-stir-welded joints of A356 aluminum alloy.

In this investigation, we conducted friction stir welding on aluminum alloy plates with different iron content. The influence of iron content on the strength of the friction-stir-welding joint was investigated by observing the microstructure, mechanical properties, and microhardness.

## 2. Materials and Methods

Scrap automobile wheel hubs were the source for recycled aluminum alloy. This study aimed to simulate the inevitable impurity of iron in the aluminum alloy recycling process by adding Al–Fe intermediate alloy. Castings of original A356 aluminum alloy without Fe added and A356 aluminum alloy with different Fe content were made using the sand casting method. There were three kinds of Fe content designed, viz., 0.2, 0.3, and 0.4 wt.%. The microstructure of the alloys with different Fe content is shown in Figure 1. It can be observed from Figure 1 that the silicon phase grows along the grain boundary of the aluminum matrix, and there are accumulations of the Fe-rich phase and silicon phase. The chemical composition of the alloys mentioned above is shown in Table 1.

The castings were machined to rectangular plates of 100 mm × 40 mm × 3.5 mm. Square butt joint configuration, as shown in Figure 2, was prepared to fabricate FSW joints. In this investigation, plates from the same casting were self-welded, which means the chemical composition of each two plates for a single weld was the same. The welding equipment used in this study is FSW-LM-AM16-2D-type Gantry Model Friction Stir Welding Machine. The stirring friction welding tool, as shown in Figure 3, has a shoulder diameter of 12.0 mm and a pin diameter of 3.0 mm. During welding, the stirring head is kept perpendicular to the sample and rotates counterclockwise. The initial joint configuration was obtained by securing the plates in position using mechanical clamps. Non-consumable tool made of H13 hot-work die steel was used to fabricate the joints. 

During the FSW process, heat was generated through the combined action of stirring and friction, causing the material to flow around the tool pin. The rotation speed and welding speed are critical factors that influence the amount of heat generated, consequently impacting the quality of the weld joint. Inadequate or excessive heat in the stirring zone can result in defects such as pinholes, tunnels, cavities, and cracks in the friction weld joint. Therefore, the selection of appropriate process parameters for FSW is imperative to ensure the production of high-quality welded joints. A study conducted by Jayaraman et al. [22] demonstrated that A356 aluminum alloy joints manufactured using process parameters of tool rotation speed of 1000 r/min, welding speed of 75 mm/min, and axial force of 5 kN exhibited higher tensile strength compared to other joints, reaching approximately 160 MPa. Dwivedi [23] investigated the influence of axial force, stirring speed, and welding speed on the quality of welded joints of A356 and C355 aluminum alloys. The results showed that a maximum tensile strength of the alloy can be achieved at an axial force of 3 kN, tool rotation speed of 900 r/min, and welding speed of 75 mm/min. In this investigation, we chose rotation speed of 1000 r/min, welding speed of 100 mm/min, and axial force of 4 kN as the welding process parameters.

The specimens used for microstructure observation were taken from welded joint. The specimens were polished with sandpaper and diamond polishing paste and then corroded with 0.5 wt% HF solution. The microstructure was observed using Zeiss Axio Vert.A1 optical microscope, and the SEM images were taken using TESCAN tungsten filament scanning electron microscope. Microhardness test was conducted as well on the cross-section of the welded joint. The Vickers hardness tester with a 100 gf load was used to take points every 0.5 mm for microhardness test. Tensile tests were carried out in a CMT5105-type testing machine. Tensile specimens were machined in the transverse direction from the welded joints and three specimens of each alloy were taken for testing. Dimensions of tensile specimen are shown in Figure 3.

## 3. Results and Discussion

### 3.1. Macrostructure

Figure 4 demonstrates the macrostructural characteristics of friction-stir-welded joints in A356 aluminum alloy, considering varying Fe contents. Upon examination, it is evident that the A356 aluminum alloy plates, featuring three distinct Fe contents, exhibit no discernible welding flaws. This observation intimates that the experiment’s selected process parameters were indeed suitable for the task. Furthermore, Figure 5 reveals that the welded joint possessing 0.3 wt.% Fe content displays the most refined weld surface. Conversely, a welded joint with 0.2 wt.% Fe content exhibits the coarsest surface morphology. Consequently, it can be deduced that the iron content serves as a crucial determinant in establishing the quality of the welded joint.

### 3.2. Microstructure

Figure 5 depicts the microstructural variations observed in the welded joints as a function of distinct Fe content values. A thorough examination reveals the presence of three primary regions within the friction-stir-welded joint, specifically: the weld nugget zone (NZ), the heat-affected zone (HAZ), and the thermo-chemically affected zone (TMAZ).

The microstructure of the NZ exhibits significant differences from that of the base metal (BM). The NZ displays a much more homogeneous microstructure in comparison to the BM. The external stirring friction force induces a transformation in the dendrite structure of the alloy, converting it into finer particles that disperse throughout the entire NZ. The material within the FSW zones undergoes intense stirring and mixing, resulting in the breakup of coarse, acicular Si particles, Chinese-character-shaped iron-rich phase particles, and the dendritic structure. This process leads to a homogeneous distribution of particles throughout the aluminum alloy matrix. Additionally, this region undergoes dynamic recrystallization, culminating in the formation of fine equiaxed grains. Cavaliere et al. [24] also reported the occurrence of plastic deformation and dynamic recrystallization as mechanisms leading to grain refinement in friction-stir-processed AA 7075 alloy. Additionally, Surekha et al. [25] also documented the observation of microstructure refinement in their study, whereby the phenomenon of grain refinement is attributed to the occurrence of dynamic recrystallization. The TMAZ is located between the NZ and the HAZ. Under the stirring action of the pin, the TMAZ also experiences frictional heat generated and undergoes significant plastic deformation but to a lesser extent compared to the NZ. Hence, dynamic recrystallization does not occur in this region, instead, the grains appear significantly elongated. Yasavol et al. [26] also observed the formation of the stir zone as a result of plastic deformation, while the formation of the thermo-mechanically affected zone was attributed to the shear direction exerted by the friction stir processing tool. In contrast to the NZ and TMAZ, the HAZ does not undergo plastic deformation. It only experiences the heat cycles associated with welding, resulting in grain coarsening. In comparison to the BM, the HAZ demonstrates a larger grain size due to the heat generated by the friction stirring, resulting in an elevation in temperature that causes the coarsening and growth of the grains. The exterior portion of the specimen (BM), which remains untouched by the FSW tool, presents with the largest grain size, akin to that found in the unprocessed alloys. Closer to the stir zone, a reduction in grain size is noted, with the smallest grain size being found at the stir zone.

SEM images of the welded joints with different Fe contents in the HAZ, NZ, and TMAZ are shown in Figure 6. Figure 7 shows the surface scanning results for Si and Fe elements, as shown in Figure 6c. Table 2 represents the point scanning results for the four points in Figure 6. The region shown in Figure 6 corresponds to the NZ of the welded joint. It is evident that eutectic silicon phases, appearing granular or in small blocks, are uniformly distributed on the aluminum matrix. The area illustrated in Figure 6 depicts the HAZ of the welded joint, where the eutectic silicon phases manifest elongated needle-like and dendritic structures, resembling the eutectic silicon structure of the base material prior to welding. The eutectic silicon grain size in the HAZ is significantly larger than in the NZ, indicating that eutectic silicon phases in the NZ undergo tearing and breaking under the action of friction stir, resulting in the current fine and uniform distribution in the NZ. Comparative analysis between Figure 6a,d,g reveals that when the iron content is 0.3 wt.%, the improvement in the microstructure of the alloy’s eutectic silicon phase is most pronounced. The distribution of eutectic silicon phases in the NZ is also the most uniform. Additionally, the alloy with 0.3 wt.% Fe content exhibits the smallest overall size of eutectic silicon in the NZ. Surface scanning results from Figure 7 show that Fe elements distribute along the grain boundaries of the aluminum alloy matrix, indicating the growth and gathering of Fe phases along the grain boundaries. In the NZ of the welded joint, no obvious iron-rich phase structure is observed. This may suggest that, under the mechanical action of friction stir and the thermal influence, the rich iron phases in the form of Chinese-character-shaped π-AlFeSi and coarse needle-like β-AlFeSi are broken into fine particles, distributed around the grain boundaries of the aluminum matrix. The modification in the morphology of the rich iron phase lowers its cleavage effect in the aluminum matrix, enhancing the overall mechanical properties of this region. 

### 3.3. Microhardness

Figure 8 illustrates the variation in microhardness across the welded joint perpendicular to the weld seam. The graph reveals a distinctive W-shaped distribution pattern of microhardness. Specifically, there is a notable decline in microhardness from the HAZ to the TMAZ. Conversely, there is a notable increase in microhardness from the TMAZ to the NZ. The region where the HAZ and TMAZ meet exhibits the lowest microhardness. This can be attributed to grain coarsening in the HAZ resulting from thermal cycles. Meanwhile, the TMAZ experiences partial dissolution, coalescence of second phases, and elongation of grains due to frictional heat and plastic deformation. Consequently, both these regions demonstrate significant softening effects, leading to a considerable decrease in microhardness. Conversely, the microhardness in the transition zone between the HAZ and NZ undergoes a notable increase. This result is due to the dissolution of second phase particles in the NZ, along with the ensuing dispersed precipitation of new second phase particles that form an equiaxed grain structure during the process of dynamic recrystallization. As a result, compared to the HAZ and TMAZ, there is an observed increment in microhardness. In a study conducted by Mishra et al. [27] on friction-stir-processed 2507 super duplex stainless steel, an increase in the microhardness of the processed material was attributed to the phenomenon of grain refinement. Similar findings were reported by Chainarong et al. [28] and Behnagh et al. [29], who observed an enhancement in hardness in friction-stir-processed aluminum alloy due to the induction of grain refinement.

With an increase in Fe content, the microhardness of the alloy exhibits an overall upward trend. The highest microhardness value in NZ is observed at 0.3 wt.% Fe, attaining a peak value of 73.8 HV.

### 3.4. Tensile Properties

Figure 9 illustrates a comparative analysis of the tensile properties of base materials with different Fe content and stir-friction-welded alloys. For the base material, an increase in Fe content generally results in a decrease in the alloy’s tensile strength (TS), yield strength (YS), and elongation (EL). As A356 alloy is a brittle material with no distinct yield stage, this study defines the engineering stress at 0.2% plastic deformation as the YS. In the base material alloy without added iron, the TS and YS of the tensile specimen are approximately 164 MPa and 88 MPa, respectively, with an EL of about 7.7%. In the 0.2 wt.% Fe alloy, the tensile performance of the specimen significantly decreases, with the TS and YS dropping to around 153 MPa and 82 MPa, respectively, and the alloy’s EL decreasing to approximately 5.8%. When the Fe content in the alloy increases to 0.3 wt.%, the alloy’s TS and YS are 149 MPa and 75 MPa, respectively, with an EL of around 5.6%. With an increase in the alloy’s Fe content to 0.4 wt.%, the TS and YS are approximately 150 MPa and 73 MPa, respectively, with an EL of around 5.5%. Compared to the base metal, the 0.2 wt.% Fe alloy experiences a decrease of 6.7% and 14.8% in TS and YS, respectively, along with a 24.7% reduction in EL. The 0.3 wt.% Fe alloy sees a decrease of 9.1% and 14.8% in TS and YS, respectively, with a 27.2% reduction in EL. Meanwhile, the 0.4 wt.% Fe alloy experiences a decrease of 8.5% and 17.0% in TS and YS, respectively, along with a 28.6% reduction in EL. The decrease in the tensile properties of the alloy can be attributed to the detrimental effects imposed by iron on the overall performance of the alloy. Previous research has also demonstrated that an increase in Fe content results in the presence of additional iron-rich phases, which adversely affect the mechanical properties of the alloy [4,30]. The tensile strength of welded joints is generally lower than that of base metals. This may be because with friction stir welding, the bonding strength of two aluminum alloy plates does not reach the strength between the inside of the original alloy.

The TS of the welded joints is generally lower than that of the base metal. For the welded joint with an Fe content of 0.2 wt.%, the TS is 148 MPa, representing a 2.6% decrease compared to the base metal. The welded joint with an Fe content of 0.3 wt.% has a TS of 148 MPa, which is comparable to the strength of the base metal. In the case of the welded joint with an Fe content of 0.4 wt.%, the TS is 149 MPa, indicating a 1.3% decrease compared to the base metal. With the increase in Fe content, the tensile strength of the friction-stir-welding joint first increases and then decreases. But there is no significant difference in TS between the welded joints after friction stir welding and the base metal. The YS of the welded joints is generally higher than that of the base metal. For the welded joint with an Fe content of 0.2 wt.%, the YS is 92 MPa, representing a 12.2% increase compared to the base metal. The welded joint with an Fe content of 0.3 wt.% has a YS of 90 MPa, indicating a 21.6% increase compared to the base metal. In the case of the welded joint with an Fe content of 0.4 wt.%, the YS is 87 MPa, showing a 17.6% increase compared to the base metal. The EL of the welded joints is not significantly different from that of the base metal. For the welded joint with an Fe content of 0.2 wt.%, the EL is 5.8%, representing a 15.5% decrease compared to the base metal. The welded joint with an Fe content of 0.3 wt.% has an EL of 5.6%, which is comparable to the base metal’s elongation. In the case of the welded joint with an Fe content of 0.4 wt.%, the EL is 5.5%, indicating a 15.6% decrease compared to the base metal. The YS of welded joints exhibits a negative correlation with the Fe content, whereas the opposite trend is observed in terms of the increase in EL. The YS of welded joints typically surpasses that of the base metal. This phenomenon can be attributed to the impact of grain refinement resulting from frictional heat and mechanical deformation, which enhance the YS of the welded joint.

In summary, the welded joint with an Fe content of 0.3 wt.% exhibits the best overall tensile performance, with the highest TS of 153 MPa, a good YS of 90 MPa, and a favorable EL of 5.7%.

### 3.5. Fracture Morphology

The tensile specimen fracture surfaces of welded joints with different iron contents are shown in Figure 10. It can be clearly seen that a dimple morphology is present, indicating quasi-cleavage fracture. This is because the α-Al matrix itself is relatively soft. During one-dimensional quasi-static tensile stress, a portion of the matrix will form tough dimples, while another portion of the matrix—due to the presence of a rich iron phase in the alloy—causes the alloy to cleave the α-Al matrix during the stretching process. Stress concentration at the tip of the rich iron phase leads to the formation of a crack source, which ultimately causes the specimen to exhibit a quasi-cleavage surface. This surface features a mixed fracture morphology containing both tough dimples and tearing edges on the fracture surface. The fracture of the tensile specimens occurred in the HAZ of the welded joint. The HAZ is greatly influenced by the heat generated during the friction stir process, resulting in larger grain sizes. This leads to larger and deeper dimples in the fracture process of the welded joint compared to the original alloy. Sharma et al. [19] also found that for as-cast A356 samples, crack initiation was usually associated with casting defects and Fatigue cracks propagated with significant amounts of crack branching and deflection. Additionally, for FSP samples, the fracture surfaces were almost perpendicular to the longitudinal axis and the crack propagated primarily along the interface between the silicon particles and the aluminum matrix and is characterized by the formation of dimples on the fracture surface. From Figure 10 circled in red, it can be observed that there are pores left by the casting process on the tensile fracture surface, with larger ones reaching several hundred micrometers and smaller ones only a few micrometers. From Figure 10b, secondary cracks and casting shrinkage porosity can be observed on the surface of the notch. 

## 4. Conclusions

In this study, three distinct A356 aluminum alloys featuring varying Fe content were meticulously designed to facilitate the creation of FSW joints. The primary objective was to scrutinize the influence of Fe content on the tensile strength of FSW joints, employing a comprehensive approach encompassing macrostructure analysis, microstructure analysis, mechanical property tests, and microhardness tests. The derived conclusions are outlined as follows:Among the three fabricated joints, the variant with an Fe content of 0.3 wt.% showcased the most favorable tensile performance. It exhibited the highest tensile strength of 153 MPa, commendable yield strength of 90 MPa, and a satisfactory elongation of 5.7%;During friction stirring, plastic deformation and dynamic recrystallization were identified as the primary mechanisms responsible for grain refinement in the NZ of the welded joint. Additionally, the fragmentation of eutectic silicon phases and enriched iron phases made substantial contributions to improving the tensile properties of the joint;The hardness distribution at the welded joint manifested a distinctive W-shaped pattern, centered around the weld seam. Notably, the NZ exhibited the highest microhardness, with the peak value occurring at an Fe content of 0.3 wt.%;The critical factors attributing to the superior tensile properties of the joints were identified as follows: a defect-free weld region, elevated hardness in the weld region, and the presence of very fine, uniformly distributed eutectic Si particles in the weld region.

This study elucidates the intricate relationships between Fe content variations and the resultant mechanical characteristics, shedding light on the nuanced factors influencing the tensile strength of FSW joints in recycled aluminum alloys.

## Figures and Tables

**Figure 1 materials-17-00064-f001:**
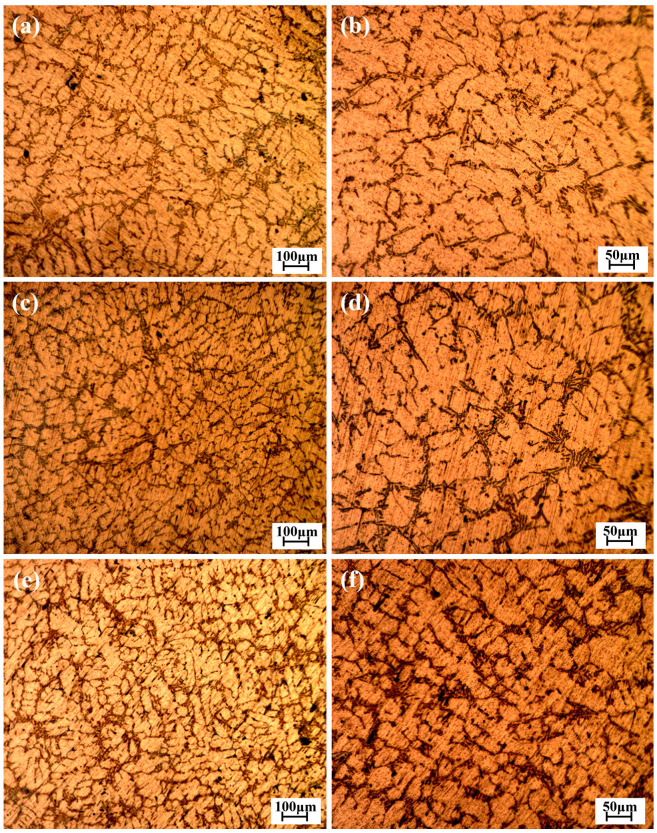
Microstructure of the alloys with different Fe content: (**a**,**b**) 0.2 wt.% Fe; (**c**,**d**) 0.3 wt.% Fe; (**e**,**f**) 0.4 wt.% Fe.

**Figure 2 materials-17-00064-f002:**
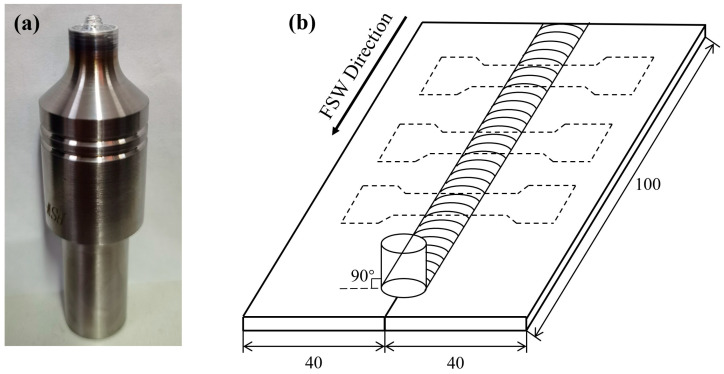
(**a**) FSW tool used in the experiment (**b**) Scheme of welding and extraction of tensile specimens.

**Figure 3 materials-17-00064-f003:**
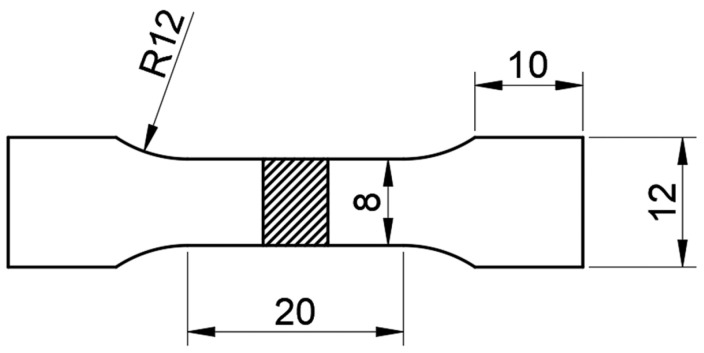
Dimensions of tensile specimen (unit: mm).

**Figure 4 materials-17-00064-f004:**
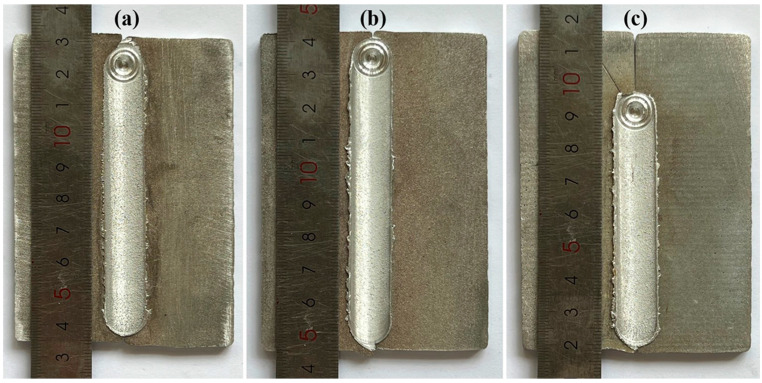
Macrostructure of the welded joint: (**a**) 0.2 wt.% Fe; (**b**) 0.3 wt.% Fe; (**c**) 0.4 wt.% Fe.

**Figure 5 materials-17-00064-f005:**
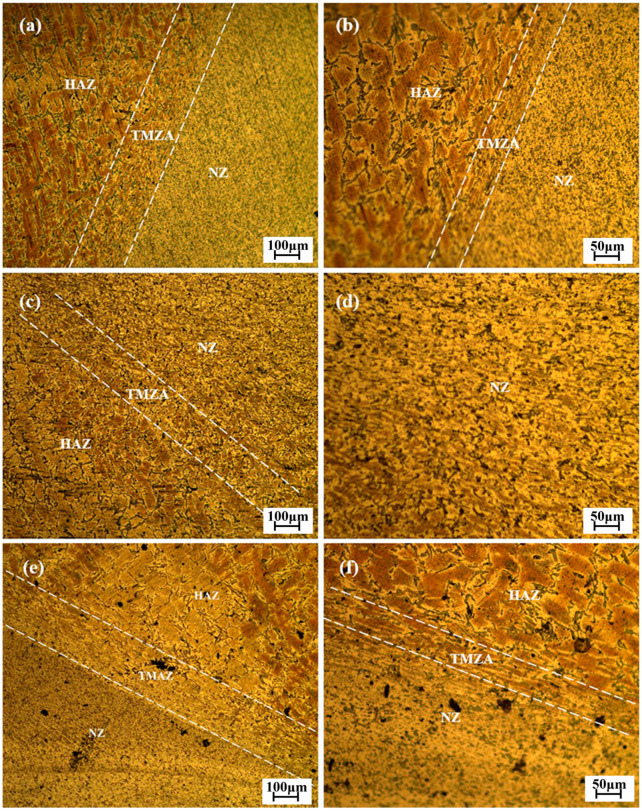
Metallographic microstructure of the welded joint: (**a**,**b**) 0.2 wt.% Fe; (**c**,**d**) 0.3 wt.% Fe; (**e**,**f**) 0.4 wt.% Fe.

**Figure 6 materials-17-00064-f006:**
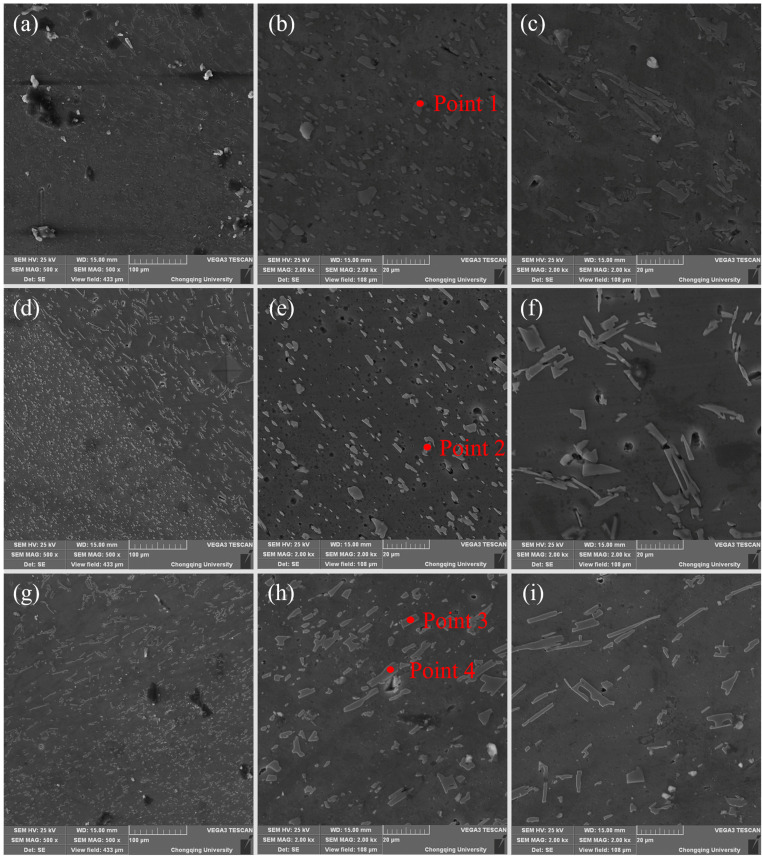
Microstructure of the welded joint: (**a**–**c**) 0.2 wt.% Fe; (**d**–**f**) 0.3 wt.% Fe; (**g**–**i**) 0.4 wt.% Fe.

**Figure 7 materials-17-00064-f007:**
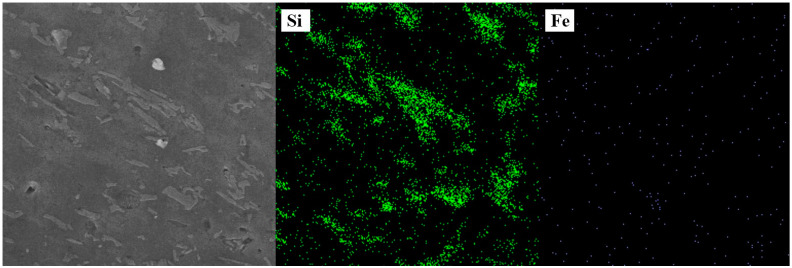
Si and Fe elements map scanning results of Figure 6c.

**Figure 8 materials-17-00064-f008:**
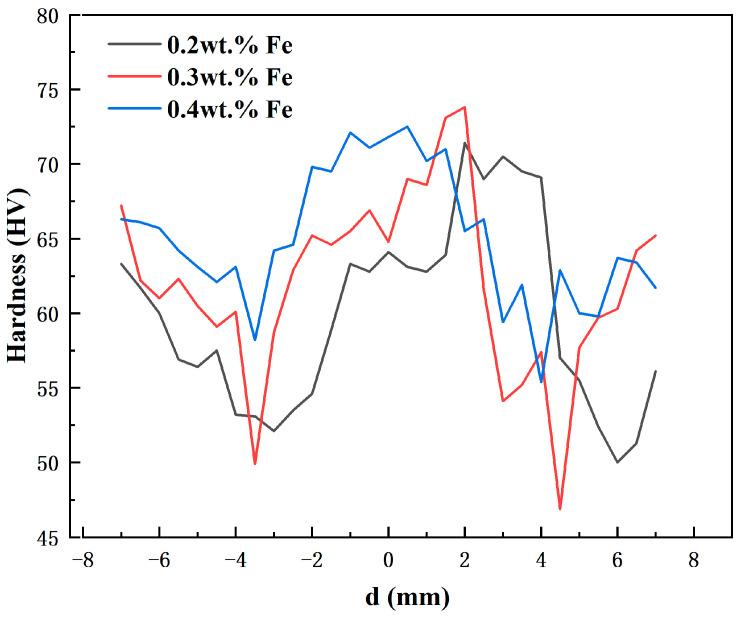
Microhardness curve of the welded joint with different Fe content.

**Figure 9 materials-17-00064-f009:**
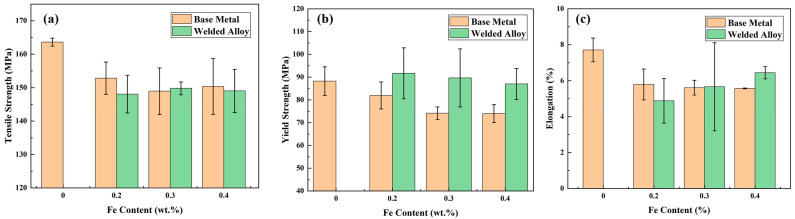
Tensile properties of the base metal and welded joint with different Fe content: (**a**) Tensile strength; (**b**) Yield Strength; (**c**) Elongation.

**Figure 10 materials-17-00064-f010:**
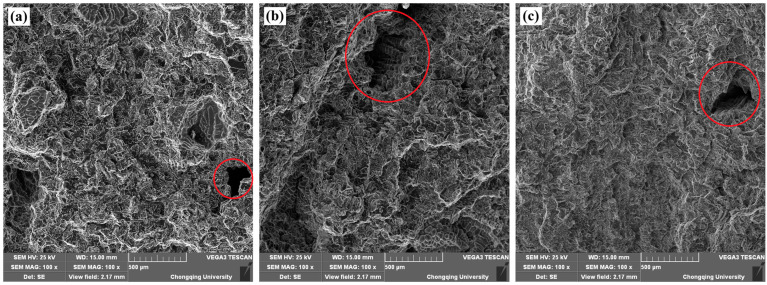
SEM fractographs showing tensile fracture surfaces of the welded joint: (**a**) 0.2 wt.% Fe; (**b**) 0.3 wt.% Fe; (**c**) 0.4 wt.% Fe.

**Table 1 materials-17-00064-t001:** Chemical composition of the alloys (wt.%).

Alloy	Si	Mg	Fe	Ti	Al
Original A356	7.06	0.42	0.11	0.12	Bal.
A356 + 0.2 wt.%Fe	6.86	0.36	0.18	0.12	Bal.
A356 + 0.3 wt.%Fe	6.75	0.39	0.32	0.14	Bal.
A356 + 0.4 wt.%Fe	6.52	0.32	0.37	0.14	Bal.

**Table 2 materials-17-00064-t002:** Point scanning results of Figure 6b,e,h (wt.%).

Point	Si	Fe	Ti	Zn	Mn	Al
Point 1	55.47	0.06	0.11	0.02	-	Bal.
Point 2	63.85	-	0.09	0.13	-	Bal.
Point 3	71.73	0.05	0.03	-	0.16	Bal.
Point 4	74.55	0.07	0.05	0.07	-	Bal.

## Data Availability

The data presented in this study are available on request from the corresponding author.

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
