# Peer review of "The Effect of Fe Content on the Microstructure and Tensile Properties of Friction-Stir-Welded Joints in Recycled Cast Aluminum Alloy"

_materials, 2023, doi:10.3390/ma17010064_

Round 1

Reviewer 1 Report

Comments and Suggestions for Authors

The application of the tested alloy was not specified. Why was Fe added if this addition reduces the mechanical parameters of the material?

The geometry and dimensions of the tools and how the tool was oriented - perpendicular or at an angle to the sample - were not provided.

Why were these welding parameters chosen?

When describing the Vickers hardness measurement, the pressure given is probably too high.

Reviewer 2 Report

Comments and Suggestions for Authors

Dear Authors,

The subject of the paper is interesting and presents valuable results in the field of aluminum casting and especially in the field of welding processes of aluminum alloys.

There are some corrections that you must do, in order to publish the article, namely:

-          - In the materials and methods section you didn’t mention anything about recycled aluminum;

-         -  Please refer to hardness in all the manuscript and also in the abstract (3 kg force is for hardness not for microhardness);

-         -  For a better visibility please add a background at the scale of the microstructure (figure 1 and 5);

-          -In figure 10 appear the images a-c and d-f not, according to the caption “Figure 10. SEM fractographs showing tensile fracture surfaces of the welded joint: (a, b) 0.2wt.% Fe (c, d) 0.3 wt.% Fe (e, f) 0.4 wt.% Fe “

-        -  Please specify how influence the FSW process the fracture morphology compared with base material

Best regards!
